# The Sustainable Development of Forest Food

**Weilung Huang [1] , Si Chen [2],\*, Xiaomei Zhang [2] and Xuemeng Zhao [1]**

1   School of Finance and Trade, Wenzhou Business College, Wenzhou 325035, China
2   School of Management, Wenzhou Business College, Wenzhou 325035, China
\*   Correspondence: 20180329@wzbc.edu.cn

**Abstract:** This paper aims to study the sustainable development of forest food by exploring the input–output relationship of forest food value chains (FFVC) and its mediating effect on the integrity and agglomeration of FFVC. Through a literature review and interviews with experts, this paper included measurement variables, such as FFVC's input, output, integrity, and agglomeration, and used PLS-SEM to study their relationships and the mediating effects of Chinese FFVC. The results showed that first, the measurement of FFVC's integrity and agglomeration focused on FFVC's rationality, development, comparative advantages, scale, space, network, and innovation; second, there was evidence of a significant input–output relationship of FFVC; third, there was a significant mediating effect of integrity and agglomeration of FFVC, which should be included in the government's policies to promote FFVC; forth, Chinese FFVC is still at its infancy, and the government must implement FFVC sustainable development policies to promote the rationalization, upgrading, and spatial coupling of integrity and agglomeration of FFVC.

**Keywords:** forest food; value chain; industrial structure; industrial agglomeration

## 1. Introduction

With an increase in governmental promotion policies and consumers' considerations of food security and nutrition, forest food has become an important economic crop, which boosts the economic sustainability of forest value chains. Therefore, the purpose of this paper is to study the sustainable development of forest food by surveying the input–output relationship of the forest food value chain (FFVC) and the factors that affect the relationship (Zhu and Zhang, 2014 [1–4]. Based on a literature review, the purposes of FFVC policies include food security, poverty-reducing, and industrial up-gradation, which was concluded as the production and marketing plan, consumer participation, industrial chain management, and FFVC surveys at a regional and national scale [5,6]. The China Forestry Bureau [7] has recognized that FFVC was the value-added sector for the forestry industry development and was conducive to developing FFVC with regional characteristics through financial subsidies.

The current literature on forest food focuses on the nutritional knowledge of forest food [8], cultural transformation [9], nutritional benefits [10], and ecological restoration potential [11]. Recently, the nutritional benefits of forests have been widely acknowledged [12]. Forest food could improve consumers' diet quality and nutrition [13,14], and Agúndez et al. [5] and Gebauer et al. [15] found that there is a growing international interest in forest food and related products in FFVC's local diet culture and tradition, health promotion, food security, species protection, and local community development. Forest foods contribute to the diversity of human diets [16–18]. Chamberlain [19] and Nurhasan [20] thought forest products are more "natural" and healthier than food produced in agriculture.

There are more than 300 types of food, pharmaceuticals, or cosmetics that contain ingredients derived from baobab in Europe. Wiersum [21] thought that the consumers in the Netherlands tend to experience the natural and cultural identity of forest food. According to Zhao et al. [22] and Yu and Jiang [23], forest food without the use of pesticides or additives

in the production process is preferred. Schumann et al. [24], Venter and Witkowski [25], Dhillion and Gustad [26] found that human activities and the types of land used were the influencing factors of product yield from the non-timber forest in West Africa. Those human activities consisted of harvesting, seedling protection, transplanting, dispersal of seeds in the garbage, livestock grazing, and plowing, and those types of land are nature reserves, rocky outcrops, plains, fields, and village (habitation) areas. Lacuna-Richman [27] discovered that forest food was considered as a supplement to food instead of the source of income, and the most important factor in forest food consumption is the effect of the market economy.

The sustainable development of FFVC should help the sustainability of forests and environmental and economic sustainability should be considered by the government, aiming to improve the healthy development of forests and their value chains [5,24–26,28]. For the sake of the ecological and environmental benefits of forests, numerous countries have banned forest value chains from exploiting the direct economic benefits of deforestation, such as logs or wood pulp. However, in light of the sustainable development of forests, countries still encourage forest value chains to develop the indirect economic benefits of the forest, such as FFVC, eco-tourism, animal and plant breeding, and utilization. Based on previous literature, the ecological and environmental benefits of forests are becoming increasingly important, such as climate regulation, air purification, soil and water conservation, and biodiversity maintenance [29–32]. Furthermore, Jendresen and Rasmussen [1] thought the bottom wealth group has a higher frequency of forest food consumption. Durazzo et al. [33] thought forests and trees are essential in food production. Sunderland and Vasquez [34] believe that we must strike a balance between maintaining forest ecological diversity and developing forest industries. Elbakidze et al. [35] believed that forests were of environmental importance for biodiversity, the global carbon cycle, and the international food trade. Graham [36] found that the structure and diversity of forest-related ecosystems may depend on the forest itself.

The act of commercial logging of state-owned natural forests has been forbidden in Northeast China since April 2014. Such an economic development model is in line with the national industrial policy, which vigorously promotes green alternative industries such as forest food, ecological tourism, planting and breeding, and forest pharmaceutical industries. Steel et al. [37] and Asprilla-Perea et al. [38] thought forest food generates community income. Makarov [39] suggested that special attention should be paid to organizing and strengthening the cultivation of forest fruits and medicinal plants to cultivate new varieties with high yields. Shackleton et al. [40] discussed the use, management, and marketing of forest food. Kusters et al. [41] and Sunderlin et al. [42] found that forest food production could contribute to forest conservation and livelihood improvement.

According to various international and national unions and regulations, forest food refers to the edible plants or parts growing naturally in forests. Based on the statistical classification of forest food in the China Forestry Statistics Yearbook, the Chinese forest coverage rate was 22.96% by the end of 2018. The output and value of forest food were 678,600 tons and CNY 10.723 billion, respectively. Compared with 2011, the value of forest food increased by 130%, which was the most substantial increment in Chinese non-wood forest products. Specifically, forest food can be divided into edible fungi, wild vegetables, and others. [21,43,44,44–51] Recent surveys show that the average European household consumes about 60 kg of forest food annually [52].

Compared with the traditional agricultural value chain (AVC), FFVC has less input of non-natural resources in its pre-production and production stages. It is because farmers do not use genetically modified crops, fertilizers, pesticides, and herbicides out of consideration of the ecological protection of forests, or the public's recognition of the forest industry is low, meaning that the forest industry's human capital cannot be improved [53]. Traditionally, AVC can be divided into the stages of pre-production, production, processing (transportation), sales, and marketing service. To be more specific, in the pre-production stage, there are crop R&D enterprises (as breeding) and crop input enterprises (as fertilizer,

pesticide, and seeds). The production stage is filled with crop planting enterprises (or farmers). The enterprises in the processing (transport) stage are typically crop processing (sorting, cutting, mixing, refining and chemical treatment) or transportation enterprises. There are wholesale or retail companies in the sales stage, and enterprises in the marketing service stage include experiential marketing service, e-commerce service, and brand-promotion service companies [54,55]. According to the results of interviews with experts, FFVC in China lacked the stages of pre-production, production, and marketing services [56–58].

Based on a literature review and interviews with experts, FFVC may be an essential research topic of AVC, and the value-added strategies of FFVC should consider its input–output relationship and its influencing factors, whereas the reality is disappointing [59–62]. On the other hand, some works of literature suggested that the input–output relationship of AVC might be impacted by the integrity and agglomeration of FFVC [22,63–65]. Therefore, the first aim of this paper is to build a structural equation model to explore the input–output relationship of FFVC and its mediating effect on FFVC integrity and agglomeration. Second, this paper further explored the spatial dependence of FFVC's integrity and agglomeration, and the results of the spatial econometric model could be used as policy recommendations for FFVC.

Agyeman and Ochuodho [66] thought capital and labor endowments positively and significantly influenced forest industry structure. Assa [67] and AFDB/OECD/UNDP [68] argued that industrial structure and scale were the important channels between foreign direct investment and forest resource degradation. Furdychko et al. [69] and Zhang et al. [70] argued that the influencing factors of regional industrial eco-efficiency in China included environmental regulation, technological innovation, level of economic development, and industrial structure. Dasgupta and Stiglitz [71] found that industrial structure and concentration and considerable size spur inventive activity were the drivers of innovations. Teece [72] indicated that market structure, firm boundaries (the level of integration), the structure of financial markets, and formal and informal organizational structure must be recognized as significant determinants of the rate and direction of innovation. Dasgupta and Stiglitz [73] proposed a relationship between research and development expenditure and suggested that industrial structures depend on more basic ingredients, such as technology research, demand conditions, and the nature of capital markets.

The government, enterprises, and scholars could carry out marketing and promoting strategies of FFVC based on the discussion of integrity and agglomeration of FFVC. Research on FFVC integrity and agglomeration must explore its mediating effect on the input–output relationship of FFVC by associating FFVC's various economic elements in different regions, which can be represented by the proportions of the various economic factors of FFVC in different regions. From the perspective of regional economic theory, the reasons why FFVC operators are concerned about the spatial concentration or clustering of FFVC can be concluded as follows: first, the development dynamics of FFVC. FFVC's operators would like to understand and make use of their regional singularities, and the successful regional clusters of FFVC would bring performance and competitiveness into its sectors with the specific locational advantages of FFVC [74–77]; second, innovation dynamics of FFVC. FFVC's innovative factors include interactive learning and regional externalities of cultural, economic, and institutional environments [74–77].

The contributions of this paper are as follows: (1) based on the industrial economy, and ecological protection of FFVC, this paper explored the input–output relationship of FFVC so as to narrow the gap in empirical studies about AVC theory. (2) This paper also explored the relationship between the input, output, integrity, and agglomeration of FFVC, which may further enrich agricultural economic theory. (3) This paper is one of the first few studies that discusses the integrity of FFVC and the mediating effect of agglomeration in the input–output relationship of FFVC and may, therefore, provide suggestions for policy-making. (4) This paper developed a structural equation model to explore the relationships among the input, output, integrity, and agglomeration of FFVC.

## 2. The Literature Review and Hypotheses of Forest Food Value Chain

According to the literature review, FFVC should be an important research topic in forestry, but very few studies discuss FFVC and its influencing factors. This paper surveys the existing literature in China and finds no literature with the title of FFVC. Furthermore, literature with the title of forest food (forestry) was first published in 1988 (1951), with 149 (72,334) respective papers as of 14 August 2022, which a focus on the input–output relationship, structural relationship, or regional competitiveness of forest food and forestry. This paper surveys the literature in WOS and finds no literature with the title of FFVC. Moreover, literature with the title of forest food (forestry) was first published in 1917 (1951), with 768 (11,339) respective papers as of 14 August 2022, which focused on the ecology, environmental sciences and studies, zoology, and biodiversity conservation of forest food and forestry. In addition, the number of such articles tends to increase with time. However, no literature discusses the effect of FFVC's integrity and agglomeration on its input–output relationship of FFVC.

Agyeman and Ochuodho [66] investigated the optimal location of potential forest industry clusters to improve the utilization of timber in Korea. Lee and Kim [78] investigated the contribution ratio of the forestry industry to the products and services of each industry in Korea to form the input–output table of the forestry industry. Li et al. [79] used fuzzy AHP to analyze the competitiveness of forest food, and its key factors are natural endowment and labor education. Dong [80] explored the competitive advantages of the Chinese agricultural value chain (such as FFVC) by constructing a panel data model and discovered that output deviation, structural effect, labor deviation, and aggregated advantage index could be the competitiveness evaluation variables of the agricultural value chain, and the competitive advantages of agricultural value chains in different regions of China were obviously different. Wang et al. [81] discussed the rationality of the forestry value chain structure in China and found out FFVC's internal spatial correlation and external spatial correlation could be the evaluation variables of its aggregation.

However, no literature discusses the effect of FFVC's integrity and agglomeration on its input–output relationship of FFVC, and the number of such articles tends to increase with time. Zhang and Peng [82] discussed the evaluation of regional competitiveness of China's forest food by factor analysis, and the results show that the resource factor is not the decisive factor for the competitiveness of forest food. Zhou [57] employed an asymmetric Nash negotiation model to explore the benefit distribution mechanism of the agricultural value chain, and it turned out that the influencing factors were the structure and agglomeration of the agricultural value chain. Deng [83] explored the rationality of the agricultural value chain structure in Sichuan of China and found that the deviation in the output value and structural effects of the agricultural value chain could be the structure evaluation variables.

Through a literature review and experts' interviews, technological innovation has promoted the development of the forest industry [84–87]. There are increasing enterprises in the production, processing (transportation), and sales stages of Chinese FFVC. As of 2020, in the Chinese Northeast state-owned forest region, there are 108 companies involved in the production stage of FFVC, 149 in its processing (transportation) stage and 50 in its sales stage. Taking Jilin Forest Food Company and Hessen Green Foods Company as examples, the main business of Jilin Forest Food Company is the cultivation, processing, and sales of wild blueberries from the forest, and its products are blueberry wine and blueberry anthocyanin. Comparatively, Hessen Green Foods Company promotes specialty forest foods, such as black fungus, blueberries, red pine seeds, black honey, and wild vegetables.

The state-owned forest areas in Northeast China are essential for more than 50% of Chinese forest areas, and they are also the primary source of forest food in China, with their output accounting for more than 98% of the output of Chinese state-owned forest areas. In Northeast China, the state-owned forest areas mainly include the Jilin forest, Heilongjiang forest, and Daxinganling forest. This paper included the FFVC of these forest areas as an example to explore its input–output relationship and the mediating effect of integrity

and agglomeration of FFVC and further explore the spatial dependence of integrity and agglomeration of FFVC.

The input variables of FFVC (FCI) are set as the capital and labor of FFVC in any region i and any time t, and many countries have used subsidies to foster FFVC, so this paper assumes the measure variables of FCI are as follows: (1) ecological protection investment ($FCI_1$, RMB), which is measured by the government's investment in ecological construction and protection in any region i and any time t. (2) Investment ($FCI_2$, RMB), whose measurement is forestry enterprises' investment in non-wood products in any region i and any time t. (3) Employees ($FCI_3$, number), which is measured by forestry enterprises' employees (on-the-job employees and retirees) in any region i and any time t (the end of the year). (4) Wages ($FCI_4$, RMB), which is represented by forestry enterprises' total wages (on-the-job) in any region i and any time t.

On the other side, the output variables of FFVC (FCO) shall be the output and value of FFVC in any region i and any time t, and the above inputs would affect the output and value of the economic forest and non-wood forest products, so this paper assumes the measure variables of FCO are as follows: (1) FFVC output ($FCO_1$, tons), whose measurement is forestry enterprises' output of forest food in any region i and any time t. (2) FFVC output value ($FCO_2$, RMB), with its measurement being forestry enterprises' output value of forest food in any region i and any time t. (3) Forest output value ($FCO_3$, RMB), which is measured by forestry enterprises' output value of economic forest in any region i and any time t. (4) Non-wood products output ($FCO_4$, tons) and its measurement is forestry enterprises' output of non-wood products in any region i and any time t. Zhang et al. [88] analyzed the forestry input–output efficiency in Beijing by the input variables, which were measured by fixed-asset investment, employees, and afforestation area, and output variables, which consisted of the structure and output value of the forestry industry, and forest greening rate. Hao et al. [89] studied the input–output efficiency of forestry in the Heilongjiang state-owned forest area, where input variables were investment, employees, and wages, and output variables were the output value of the forest and the output of non-wood forest products. Feng et al. [90] explored the input–output relationships of forestry, with the input variables being represented by ecological construction, protection investment, and employees of forestry and the output variable being the output value of forestry.

The mediating effect of FFVC's integrity on its input–output relationship is due to FFVC's integrity variables (FSI), which affect its resource allocation efficiency and then modify its production efficiency, so this paper assumes the measure variables of FSI are as follows:

1.  structural effect ($FSI_1$), $FSI_1 = \{FSI_{i,t,1}\}_{\forall i, \forall t}$ and the measurement of $FSI_{i,t,1}$ is $FSI_{i,t,1} = FCO_{i,b,2}[(FCO_{t,2} - FCO_{b,2})/FCO_{b,2} - (N_t - N_b)/N_b]$, which is the multiple of forestry enterprises' output value of forest food in the base period (t = b) and region i ($FCO_{i,b,2}$). The increment in the equation is generated by the difference between the average growth rate of the forest food output value in all regions from the base period to time t $((FCO_{t,2} - FCO_{b,2})/FCO_{b,2})$ and the average growth rate of the output value of non-wood products, which is classified as the upper-level food of forest food, in all regions from the base period to time t $((N_t - N_b)/N_b)$; where $FCO_{t,2}$ ($FCO_{b,2}$) is the forestry enterprises' output values of forest food in all regions and time t (base period), and $N_t$ ($N_b$) is the forestry enterprises' output values of non-wood products in all regions and time t (base period). $FSI_1$ is an indicator of the contribution from the growth of the FFVC output value. The more the $FSI_1$ is, the more the contribution of the forestry enterprises' output value of forest food in region i and time t will be.
2.  Labor deviation ($FSI_2$), $FSI_2 = \{FSI_{i,t,2}\}_{\forall i, \forall t}$, and the measurement of $FSI_{i,t,2}$ is $FSI_{i,t,2} = FCO_{i,t,2}/L_{i,t} - FCO_{t,2}/L_t$, which is the difference between the labor productivity of forest food in region i and time t ($FCO_{i,t,2}/L_{i,t}$) and the labor productivity of forest food in all regions and time t ($FCO_{t,2}/L_t$); where $L_{i,t}$ ($L_t$) is the labor of forest food in region i (all regions) and time t.

3.　Output value deviation ($FSI_3$), $FSI_3 = \{FSI_{i,t,3}\}_{\forall i,\ \forall t}$, and the measurement of $FSI_{i,t,3}$ is $FSI_{i,t,3} = (N_{i,t}/L_{N,i,t} - FCO_{i,t,2}/L_{i,t})/FCO_{i,t,2}/L_{i,t}$, which represent the asymmetry generated by the difference between the labor productivity of non-wood products ($N_{i,t}/L_{N,i,t}$) and the labor productivity of forest food ($FCO_{i,t,2}/L_{i,t}$) in region I and time t, where $L_{N,i,t}$ is the labor of non-wood products in region i and time t. The more the $FSI_{i,t,2}$ is, the more the integrity of FFSI in region i and time t will be. The less the $FSI_{i,t,3}$ is, the more possibility of labor regarding FFSI in region i and time t.

There is another mediating effect of FFVC's agglomeration on its input–output relationship, which is because FFVC's agglomeration variables (FSA) affect and support each enterprise in FFVC for its commonality and complementarity, and then modify its operating efficiency. In addition, it would still learn and co-innovate with companies and institutions in the FFVC. Thus, this paper uses location entropy as a useful indicator in measuring the spatial distribution of FFVC because of its specialization in FFVC. For a regional input–output model, it is necessary to apply the location entropies with the employment base and output value base, which is a ratio of a regional FFVC's share of output value or employment to a similar share in a broader region.

This paper assumes the measure variables of FSA as (1) the location entropy of output value ($FSA_1$), $FSA_1 = \{FSA_{i,t,1}\}_{\forall i,\ \forall t}$, and the measurement of $FSA_{i,t,1}$ is $FSA_{i,t,1} = FCO_{i,t,2}/FCO_{t,2}/N_{i,t}/N_t$, which is the ratio on the output value of forest food and non-wood products in region i and time t related to that in all regions and time t, where $N_{i,t}$ is the output value of non-wood products in region i and time t. (2) For the location entropy of employment ($FSA_2$), $FSA_2 = \{FSA_{i,t,2}\}_{\forall i,\ \forall t}$, and the measurement of $FSA_{i,t,2}$ is $FSA_{i,t,2} = L_{i,t}/L_{N,i,t}/L_t/L_{N,t}$, which is the ratio of the labor of forest food and non-wood products in the relevant region i and time t to that in all regions and time t, where $L_{N,t}$ is the labor of non-wood products in all regions and time t. (3) For the spatial concentration of marketing ($FSA_3$), $FSA_2 = \{FSA_{i,t,2}\}_{\forall i,\ \forall t}$, and the measurement of $FSA_{i,t,3}$ is $FSA_{i,t,3} = M_{i,t}/M_t$, which is the ratio of the sales of forest food in region i and time t ($M_{i,t}$) to that in all regions and time t ($M_t$). (4) For the spatial concentration of enterprises ($FSA_4$), $FSA_4 = \{FSA_{i,t,4}\}_{\forall i,\ \forall t}$, and the measurement of $FSA_{i,t,4}$ is $FSA_{i,t,4} = E_{i,t}/E_t$, which is the ratio of the number of forest food enterprises in a region i and time t ($E_{i,t}$) to that in all regions and time t ($E_t$). (5) For internal spatial correlation ($FSA_5$), $FSA_5 = \{FSA_{i,t,5}\}_{\forall i,\ \forall t}$, and the measurement of $FSA_{i,t,5}$ is $FSA_{i,t,5} = \dfrac{\min\limits_{i,t}|FCOI_{t,2}-FCOI_{i,t,2}| + \rho(\max\limits_{i,t}|FCOI_{t,2}-FCOI_{i,t,2}|)}{|FCOI_{t,2}-FCOI_{i,t,2}| + \rho(\max\limits_{i,t}|FCOI_{t,2}-FCOI_{i,t,2}|)}$, where $\rho$ is the identification rate, $FCOI_{t,2} = \dfrac{FCO_{t,2}}{FCO_{b,2}}$; $FCOI_{t,2} = \dfrac{FCO_{i,t,2}}{FCO_{i,b,2}}$. For time-dependent factors of $FCO_{t,2}$ and $FCO_{i,\,t,2}$, in order to avoid the impact of the protruding shape, $FSA_{i,t,5}$ data must be processed with a value normalization, so these values can be sorted to the order of 1 after dividing by the base value. All data are corrected to the same scale and position, and then the distance of each of the data is expressed by the absolute value of the base value (the distance is positive) [91–94].

Based on the above discussion, the methodology, hypotheses and research designs are illustrated in Figure 1. The paths of the model (Hypotheses) are as follows:

$$FCO = \beta_1 FCI + \epsilon_1 \tag{1}$$

$$FCO = \beta_2 FCI + \beta_3 FSI + \epsilon_2 \tag{2}$$

$$FCO = \beta_4 FCI + \beta_5 FSA + \epsilon_3 \tag{3}$$

$$FCO = \beta_6 FCI + \beta_7 FSI + \beta_8 FSA + \epsilon_5 \tag{4}$$

where $\epsilon_{1,i}$, $\epsilon_2$, $\epsilon_3$ and $\epsilon_4$ are residual variances. Equation (1) tests the hypothesis ($H_1$) that FCI has a significant effect on FCO (FCI → FCO). Similarly, Equations (2)–(4) test the hypotheses ($H_2$, $H_3$, and $H_4$) that the mediator variables have significant effects on the dependent variables (FCI → FSI → FCO, FCI → FSA → FCO, and $FCI \overset{\rightarrow}{\rightarrow} \begin{matrix} FSI \\ FSA \end{matrix} \overset{\rightarrow}{\rightarrow} FCO$).

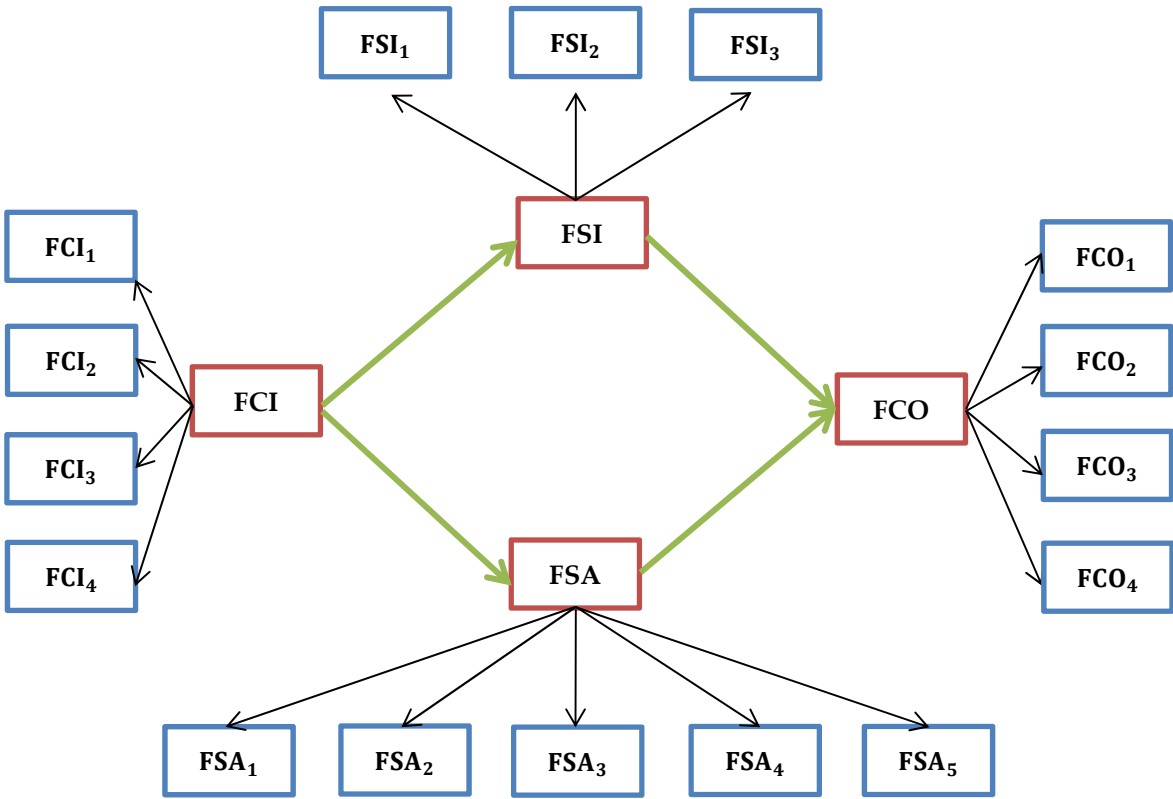

**Figure 1.** The methodology, hypothesis, and research architecture.

## 3. The Structural Equations of Forest Food Value Chain

This study uses SPSS 24.0 and Smart PLS 4.0 to solve the structural equation models by PLS-SEM (variance-based SEM), with three regions (Jilin forest area, Heilongjiang forest area, and Daxinganling forest area) and eight years (2011~2018) of valid reports from the China Forestry Bureau [44–50] and China Forestry and Grassland Bureau [43].

The descriptive statistics (see Table 1) suggest the following findings:

(1) For FCI, in three regions in 2011~2018, the Chinese government's investment in ecological construction and protection was CNY 3476.93 million on average; Chinese forestry enterprises' investment was CNY 77.31 million on average; the average number of Chinese forestry enterprises' employees was 145,861.75; Chinese forestry enterprises' total wages, on average, was CNY 25,156.55 million. From the literature review and experts' survey, the meanings of FCI's values are as follows: A. the average value of Chinese ecological construction and protection investment accounts for 23.18% of the average investment of the forestry industry, which is the highest proportion of investment, except for forest plantation investment. It means that the development purpose of Chinese forestry should be in line with the forest food industry, which promotes forest protection, conservation, and ecological balance. B. The forestry enterprises' investment in non-wood products occupies 1.23% (the lowest proportion) of the forestry industry investment due to the government budget constraints. The Chinese FFVC could be viewed as a startup industry, which is full of prospects. C. The average employee number of non-wood forest products accounts for 28.15% of the forestry industry, and it has been increasing year by year, which means that the transformation of Chinese FFVC has allowed more and more workers to switch from logging to FFVC. D. Since logging has been stopped in China, the total wages of employees in the forestry industry have decreased.

(2) For FCO, in three regions in 2011~2018, the average weight of Chinese forestry enterprises' output of forest food was 551.22 million tons; Chinese forestry enterprises' output value of forest food, on average, was CNY 1402.61 million; Chinese forestry

enterprises' output value of the economic forest was CNY 2768.98 million on average; the average weight of Chinese forestry enterprises' output of non-wood products was 1381.84 million tons. Based on the above results, the average output of forest food accounts for 39.87% of the average output of non-wood forest products, and it is the highest proportion of non-wood forest products other than fruits in China, which has increased year by year. The average output value of forest food accounts for 50.67% of the average output value of economic forests. The output value of forest food accounts for more than half of the output value of economic forests and has become a pillar industry in economic forests.

(3)  For FSI, in three regions in 2011~2018, the average value on the structure effect of Chinese FFVC was 2.68; the average amount of labor deviation of Chinese FFVC was CNY 275.62 million; the output value deviation of Chinese FFVC was CNY 3.89 million on average. From the results, the imbalance in Chinese FFVC's structure was caused by insufficient investment in the forestry industry. The higher index of Chinese FFVC's structure effect was because the forest food output value accounts for more than half of the economic forest output value. The higher labor deviation thereby could promote the balance of FFVC's structural benefits.

(4)  For FSA, in three regions in 2011~2018, the average value of location entropy of Chinese FFVC's output value was 0.90; the average value of location entropy of Chinese FFVC's employment was 2.30; the average amount of spatial concentration of Chinese FFVC's marketing was 0.05; the average amount of spatial concentration of Chinese FFVC's enterprises was 0.19; the average amount of internal spatial correlation of Chinese FFVC was 0.76. From the results, FFVC had no comparative advantage in terms of its output value and employment and had a poor spatial concentration of its enterprises and a strong internal correlation. This could be due to the broader space distribution of Chinese FFVC areas and no regional cooperation mechanism.

**Table 1.** The descriptive statistics of the studied variables.

| Variables | Measurements | Mean | S.E. | Loading |
|---|---|---|---|---|
| FCI | $FCI_1$ | 3476.93 | 2208.57 | 0.86 |
| | $FCI_2$ | 77.31 | 158.37 | 0.91 |
| | $FCI_3$ | 145,861.75 | 112,855.53 | 0.34 |
| | $FCI_4$ | 25,156.55 | 14,860.01 | 0.98 |
| FCO | $FCO_1$ | 551.22 | 594.76 | 0.96 |
| | $FCO_2$ | 1402.61 | 1512.82 | 0.96 |
| | $FCO_3$ | 2768.98 | 2482.90 | 0.98 |
| | $FCO_4$ | 1381.84 | 1579.40 | 0.85 |
| FSI | $FSI_1$ | 2.68 | 3.69 | −0.80 |
| | $FSI_2$ | 275.62 | 233.66 | 0.82 |
| | $FSI_3$ | 3.89 | 5.76 | 0.92 |
| FSA | $FSA_1$ | 0.90 | 0.28 | 0.60 |
| | $FSA_2$ | 2.30 | 2.10 | −0.74 |
| | $FSA_3$ | 0.05 | 0.04 | 0.96 |
| | $FSA_4$ | 0.19 | 0.12 | 0.92 |
| | $FSA_5$ | 0.76 | 0.15 | 0.54 |

This paper uses PLS-SEM, which applies the path weighting scheme and obtains the path coefficients of Figure 1 (see Figure 2). From Figure 2, it can be observed that most path coefficients are positive, which means that most correlations of the variables of FCI,

FCO, FSI, and FSA are positive. These results are consistent with the theory and experience that an enterprise with more FCI would consider more FSI and FSA and then produce more FCO. The variable that most significantly affects FFVC's output, the output value of the economic forest, and the non-timber forest products is the total wages of on-the-job workers. This result validates the dependence of the FFVC on the processing link; the most significant variable that affects FFVC's output value is its spatial concentration. This is consistent with the hypothesis in this paper that FFVC's spatial concentration has a significant impact on FFVC's output.

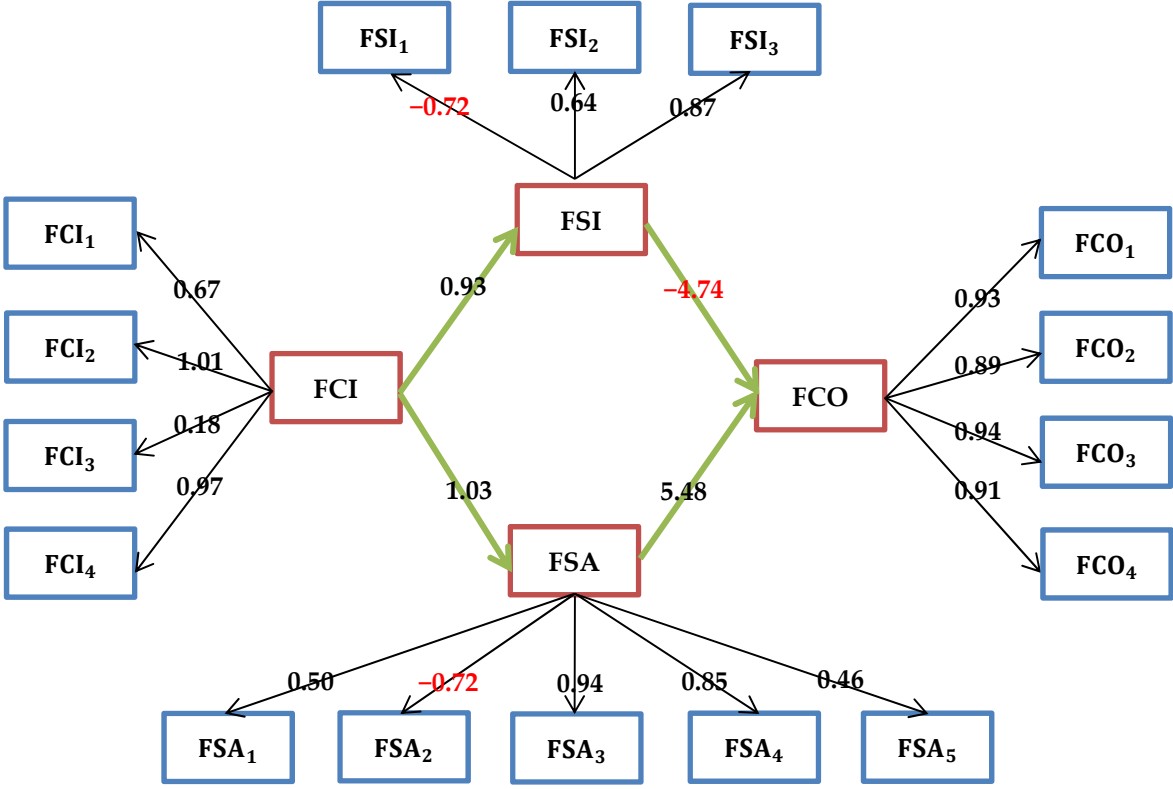

**Figure 2.** The path weighting scheme and the path coefficients of FFVC model.

Only FSI → FCO, FSI → $FSI_1$ and FSA → $FSA_2$ were negative, based on the above Figure 2. The reasons for the negative coefficient between the FSI and FCO are that FFVC was still in the rapid development stage, and the effect of FFVC's integrity on its output was not stable, which is discussed in Table 2. The negative coefficient between the FSI and $FSI_1$ was because the output value of FFVC accounts for a high proportion of the output value of economic forest products, but the lack of FFVC's investment led to a negative coefficient between FFVC's output value deviation and FFVC's integrity. The reasons for the negative coefficient between the FSA and $FSA_2$ were that there are far more workers in FFVC's processing stage than in FFVC's marketing and transportation due to the imbalance in FFVC's employment.

The reliability and convergent validity of the variables were acceptable (see Table 2). The factor loadings (loading), Cronbach's alpha ($\alpha$), rho_A, composite reliability (CR), average variance extracted (AVE), $R^2$, and Adj. $R^2$ on FCI, FCO, FSI, FSA well fitted the requirements of the analysis. For simplicity, this paper calculated the existing discriminant validity of these variables, which was, nevertheless, not listed in the table.

**Table 2.** Cronbach's alpha, rho_A, CR, AVE, $R^2$, Adj. $R^2$ on the studied variables.

| Variables | Measurements | $\alpha$ | rho_A | CR | AVE | $R^2$ | Adj. $R^2$ |
|-----------|--------------|----------|-------|-----|-----|-------|------------|
| FCI | FCI$_1$ | 0.81 | 0.95 | 0.87 | 0.66 | - | - |
|  | FCI$_2$ |  |  |  |  |  |  |
|  | FCI$_3$ |  |  |  |  |  |  |
|  | FCI$_4$ |  |  |  |  |  |  |
| FCO | FCO$_1$ | 0.96 | 0.96 | 0.97 | 0.88 | 0.91 | 0.90 |
|  | FCO$_2$ |  |  |  |  |  |  |
|  | FCO$_3$ |  |  |  |  |  |  |
|  | FCO$_4$ |  |  |  |  |  |  |
| FSI | FSI$_1$ | 0.50 | 0.88 | 0.72 | 0.59 | 0.68 | 0.66 |
|  | FSI$_2$ |  |  |  |  |  |  |
|  | FSI$_3$ |  |  |  |  |  |  |
| FSA | FSA$_1$ | −0.48 | 0.82 | 0.51 | 0.72 | 0.88 | 0.88 |
|  | FSA$_2$ |  |  |  |  |  |  |
|  | FSA$_3$ |  |  |  |  |  |  |
|  | FSA$_4$ |  |  |  |  |  |  |
|  | FSA$_5$ |  |  |  |  |  |  |

In order to carry out further study, this paper used the bootstrapping method to collate the test results and the significance of the path and mediators' path coefficients, as shown in Table 3. From the results of Tables 2 and 3, it can be observed that H$_1$, H$_2$ and H$_4$ are well supported. Only H$_3$ is not supported, but it would be statistically significant at a 90% confidence level. The results of the hypothetic tests proved that (1) there is evidence of a significant input–output relationship of FFVC, which could complement the empirical results of AVC theory. (2) There are the significant mediating effects of FFVC's integrity in the input–output relationship of FFVC, which could be evidence of the effects of many national policies. In addition, the mediating effects of FFVC's agglomeration are not significant, but these might be different with more and more empirical evidence.

**Table 3.** The significance of path and mediators' path coefficients.

|  | Mean (S.D.) |  | Mean (S.D.) |
|--|-------------|--|-------------|
| FCI $\rightarrow$ FCO | 0.91 *** (0.03) | FCI $\rightarrow$ FSI $\rightarrow$ FCO | 0.67 *** (0.10) |
| FCI $\rightarrow$ FSI | 0.84 *** (0.05) | FCI $\rightarrow$ FSA $\rightarrow$ FCO | 0.24 (0.10) |
| FCI $\rightarrow$ FSA | 0.94 *** (0.02) |  |  |
| FSI $\rightarrow$ FCO | 0.72 *** (0.11) |  |  |
| FSA $\rightarrow$ FCO | 0.68 (0.11) |  |  |

Note: *** $p < 1\%$.

## 4. Conclusions

Most countries have entirely stopped the commercial logging of natural forests. Furthermore, in order to improve economic and environmental efficiency, the wood products of forestry have been transformed into new products and services, such as forest food, ecological tourism, forest farming, and forest pharmaceuticals. As consumers are developing healthy eating habits, forest food has been viewed as a new economic growth point of forestry. Therefore, it has been suggested that policy-making should include FFVC, covering the overall value chains of forest food as an important target. However, there are few studies that discuss the input–output relationship of FFVC and the factors that

affect this relationship. Thus, this paper aimed to explore the input–output relationship of FFVC and how it was affected by the integrity and agglomeration of FFVC. The mediating effects of the integrity and agglomeration of FFVC were included, as various experts have suggested that FFVC might be improved by the policies of the integrity and agglomeration of FFVC, although little mathematical evidence is available.

This paper used the structural equation models of PLS-SEM to analyze the data of FCI, FCO, FSI, and FSA in the Jilin forest area, Heilongjiang forest area, and Daxinganling forest area over the period of 2011~2018, and SPSS was used in data analysis. The results showed that there is evidence of a significant input–output relationship of FFVC through the significant mediating effects of integrity and agglomeration.

Based on a literature review and interview with experts and the abovementioned results of FCI, FCO, FSI, FSA from PLS-SEM, some suggestions are provided. China has set a series of goals to contribute to global sustainable development, including an increase in forestry and aims to develop forestry tourism [95].

(1) Chinese FFVC is in its infancy. As for forest conservation and industrial development, the government is encouraged to employ policies, subsidies, and investment to help FFVC by extending the length of the industrial chain. The governments should firstly maximize the proportion of deep processing of forest food and increase its added value. Secondly, the governments should widen the width of the industrial chain, which could maximize the comprehensive utilization level of forest food and improve the various sub-industrial links and functions. Thirdly, the government is advised to increase the scale of the industrial chain and enlarge the scale of FFVC and enhance market competitiveness. This could be embodied in the resource allocation of forest food production, which would shift from low-efficiency (productivity) sub-industries to high-efficiency (productivity) sub-industries.

(2) Governments may enhance the rationalization and upgrading of Chinese FFVC. Strengthening the coordination ability of FFVC's sub-industries may help to increase their correlation and actively promote technological progress to improve the overall quality and efficiency of FFVC. Possible methods may include the following: A. the government relaxes the FFVC's constraints on institutions, resource market access, and infrastructures; B. the government may strengthen FFVC's correlations, such as the horizontal and vertical chain-network structure and governance mechanisms; C. the government may upgrade FFVC's enterprises in their interaction with processors, exporters and international retailers, and it is necessary to acquire new technologies, skills, and knowledge [54].

(3) Increased spatial coupling of Chinese FFVC's integrity and agglomeration would be beneficial. The government may, therefore, help FFVC to set up a forest food processing park so as to link FFVC's supply chain and share the information of production, inventory management, logistics, and distribution, which will cut costs and improve productivity. On the other side, the government is advised to establish an e-commerce platform to connect the demand chain of FFVC, and implement forest food marketing strategies according to the consumption levels in different regions, and scientifically plan FFVC's spatial layout.

(4) FFVC's related activities and industries should be developed for the sustainability of forests and forestry and these include the activities and industries of forest-based health preservation, forest-based carbon sinks, featured economic forests, bamboo and rattan, seedlings and flowers, wild animals and plant breeding and utilization, and forest-based Internet of Things. Ma and Zheng [96] thought the developments of the under-forest economy are important in the sustainability of forestry. Chu and Zhang [4] used the theory of ecology-industry symbiosis to analyze the inner and outer interaction mechanism between forest ecological security and forest food security. Its results show that there is a mutually reinforcing relationship between forest ecological security and forest food security.

A future study could focus on the mediating effects of FFVC's agglomeration and integrity on its input–output relationship. As can be observed in Table 1, the standard errors of the variables regarding FFVC's agglomeration and integrity are significant compared with their means, which may be caused by the differences across different regions. For specific suggestions of FFVC in different regions, empirical analysis in future studies could be conducted through a spatial econometric model to explore the differences across FFVC's structure level and spatial concentration in the three regions of the Northeast state-owned forest region, including the Jilin forest district, Heilongjiang forest district, and Daxinganling forest district. As a result, the interdependent spatial dependencies of FFVC's structure level and spatial concentration in the three regions may be supported. Spatial autocorrelation of FFVC refers to the implicit correlation between the same economic variables of FFVC in different regions. Measuring the interdependence of the same economic variable of FFVC at different locations is called the statistics of spatial autocorrelation.

For spatial dependence of FFVC, a stronger spatial dependence of FFVC's economic variables in a region is related to closer cooperation with neighboring areas and may promote FFVC's development. Otherwise, independent development can be confirmed. According to the strength of the spatial dependence of FFVC's economic variables between the local and neighboring areas, the FFVC's cooperation direction with the neighboring government may be studied. Therefore, it is necessary to explore the spatial dependence of FFVC's economic variables to explore their correlations [97]. In this paper, the spatial autocorrelation test of Moran's I index is used to measure the spatial dependence of FFVC's agglomeration and integrity in the three regions. Moran's I index is a measure of spatial autocorrelation developed and used by Alvioli et al. [98], Helbich et al. [99], Grieve [100], Getis and Ord [101], Li et al. [102], and Moran [103]. This index assumes that spatial autocorrelation is characterized by a correlation in a signal among nearby locations in space.

Future studies could focus on the issues of FFVC, such as its economic–environmental relationships (including forest ecosystem services, forest land ownerships and forestry's efficiency), carbon sequestration, landscape dynamics, routing and scheduling of transportation, livelihood resilience, social embeddedness, food sovereignty, and affirmative policies that include women [104–124]. As Johansson et al. [125] discussed, women professionals face challenges in the male-dominated field of forestry. Jefferson and Adhikari [126] thought governments should promote food sovereignty and protect plant varieties as intellectual property. Moreno-Calles et al. [127] thought the development of forestry should consider forestry and agricultural diversity, soil, water, cultural richness, food sovereignty, and sustainable ecosystem management. Taylor and Cheng [128] thought the social embeddedness of forestry could promote democracy, local livelihoods, and sustainable forest ecosystems.

Future studies should pay attention to significant socio-economic phenomena such as COVID-19, which might be represented in statistical yearbooks of 2020–2022 as a critical factor for policy and lifestyle changes. The limitation of data in this paper is due to the fact that the data of the research object in the China Forestry Statistical Yearbook selected in this study were only consistent from 2011 to 2018, and there are some missing data after 2018.

**Author Contributions:** Conceptualization, W.H. and S.C.; methodology, W.H.; software, W.H.; validation, W.H., S.C. and X.Z. (Xiaomei Zhang); formal analysis, X.Z. (Xuemeng Zhao); investigation, W.H.; resources, S.C.; data curation, S.C.; writing—original draft preparation, W.H. and S.C.; writing—review and editing, W.H., S.C., X.Z. (Xiaomei Zhang) and X.Z. (Xuemeng Zhao); visualization, X.Z. (Xuemeng Zhao); supervision, X.Z. (Xiaomei Zhang); project administration, X.Z. (Xiaomei Zhang); funding acquisition, W.H. All authors have read and agreed to the published version of the manuscript.

**Funding:** This research was supported by the Humanities and Social Science Projects of Chinese Ministry of Education (grant No. 21YJA790027).

**Informed Consent Statement:** Not applicable.

**Conflicts of Interest:** The authors declare no conflict of interest.

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
