# Peer review of "The Sustainable Development of Forest Food"

_sustainability, doi:10.3390/su142013092_

Round 1

Reviewer 1 Report

The structure of the introduction is not clear and the policy recomendation don´t take into consideration all aspect of forestry.

In my opinion there is a severe lack in non considering the social  environment. In particular I expect that issues like social embeddedness of foresrty, food sovereignty, affirmative policy to include women in forestry management were considered.

References in introduction should be improved.

The paragraphs from line 96 to 134 need to be reformulated, it is very hard to follow the discourse.

In line 127 there is a -first- not followed by a secondo and so on.

In line 229 there is a typo and the -i- is written -I-, please correct.

Author Response

Response to Reviewer 1 Comments

Point 1: The structure of the introduction is not clear and the policy recommendation don´t take into consideration all aspect of forestry.

Response 1:

Thank you very much for indicating the structure of the introduction. We have modified the introduction. As ”

  1. Introduction

With an increase in governmental promotion policies and consumers' considerations of food security and nutrition, forest food has become an important economic crop, which boosts the economic sustainability of forest value chains. Therefore, the purpose of this paper is to study the sustainable development of forest food by surveying the input-output relationship of the forest food value chain (FFVC) and the factors that affect the relationship (Zhu and Zhang, 2014; Jiang and Xu, 2016; Zhang et al., 2017; Jendresen and Rasmussen, 2022). Based on a literature review, the purposes of FFVC policies include food security, poverty-reducing, and industrial up-gradation, which was concluded as the production and marketing plan, consumer participation, industrial chain management, and FFVC surveys at a regional and national scale (Agúndez et al., 2018; Food and Agriculture Organization of the United Nations, 2016). China Forestry Bureau has recognized that FFVC was the value-added sector for the forestry industry development and was conducive to developing FFVC with regional characteristics through financial subsidies (China Forestry Bureau, 2017b).

The current literature on forest food focuses on the nutritional knowledge of forest food (Bukowski and Munsell, 2018), cultural transformation (Wartman et al., 2018), nutritional benefits (Nytofte and Henriksen, 2019), and ecological restoration potential (Park and Higgs, 2018). Recently, the nutritional benefits of forests have been widely acknowledged (Gergel et al., 2020). Forest food could improve consumers’ diet quality and nutrition (Albrecht and Wiek, 2021; Tata et al., 2019), and Agúndez et al. (2018) and Gebauer et al. (2016) found there had been growing international interest in forest food and related products in FFVC’s local diet culture and tradition, health promotion, food security, species protection, and local community development. Forest foods contribute to the diversity of human diets (Hall et al., 2019; Rasmussen et al., 2019, 2020). Chamberlain (2020) and Nurhasan (2022) thought forest products are more "natural" and healthier than food produced in agriculture.

There are more than 300 types of food, pharmaceuticals, or cosmetics containing ingredients derived from baobab in Europe. Wiersum (2017) thought that the consumers in the Netherlands tend to experience the natural and cultural identity of forest food. According to Zhao et al. (2015) and Yu and Jiang (2012), forest food could be appreciated with no pesticides and no additives in the production process. Schumann et al. (2011), Venter and Witkowski (2011), Dhillion and Gustad (2004) found that human activities and the used types of land were the influencing factors of product yield from the non-timber forest in West Africa. Those human activities consisted of harvesting, seedling protection, transplanting, dispersal of seeds in the garbage, livestock grazing, and plowing, and those used types of land are nature reserves, rocky outcrops, plains, fields, and village (habitation) areas. Lacuna-Richman (2006) discovered that forest food was considered as a supplement to food instead of the source of income, and the most important factor in forest food consumption shall be the effect of the market economy.

The sustainable development of FFVC should help the sustainability of forests, for environmental and economic sustainability should be considered for the government aiming to improve the healthy development of forests and their value chains (Dalei and Gupt, 2019; Agúndez et al., 2018; Schumann et al., 2011; Venter and Witkowski, 2011; Dhillion and Gustad, 2004). For the sake of the ecological and environmental benefits of forests, numerous countries have banned forest value chains from exploiting direct economic benefits of deforestation, such as logs or wood pulp. However, in the light of sustainable development of forests, countries still encourage forest value chains to develop the indirect economic benefits of the forest, such as FFVC, eco-tourism, animal and plant breeding, and utilization. Based on previous literature, the ecological and environmental benefits of forests are becoming increasingly important, such as climate regulation, air purification, soil and water conservation, and biodiversity maintenance (Molander et al., 2022; Liu et al., 2022; Villamor et al., 2022; Andrea, 2022). Furthermore, Jendresen and Rasmussen (2022) thought the bottom wealth group has a higher frequency of forest food consumption. Durazzo et al. (2020) thought forests and trees are essential in food production. Sunderland and Vasquez (2020) believe that we must strike a balance between maintaining forest ecological diversity and developing forest industries. Elbakidze et al. (2007) believed that forests were of environmental importance for biodiversity, the global carbon cycle, and the international food trade. Graham (2004) found that structure and diversity of forest-related ecosystems may depend on the forest itself.

The commercial logging of state-owned natural forests has been forbidden in northeast China since April 2014. Such an economic development model is in line with the national industrial policy, which vigorously promotes green alternative industries such as forest food, ecological tourism, planting and breeding, and forest pharmaceutical industries. Shackleton et al. (2011) discussed the use, management, and marketing of forest food. Kusters et al. (2006) and Sunderlin et al. (2005) found that forest food production could contribute to forest conservation and livelihood improvement. Steel et al. (2022) and Asprilla-Perea et al. (2019) thought forest food generates community income. Makarov (2019) suggested that special attention should be paid to organizing and strengthening the cultivation of forest fruits and medicinal plants to cultivate new varieties with high yields.

According to various international and national unions and regulations, forest food refers to the edible plants or parts growing naturally in forests. Based on the statistical classification of forest food in the China Forestry Statistics Yearbook, the Chinese forest coverage rate was 22.96% by the end of 2018. The output and value of forest food were 678,600 tons and RMB 10.723 billion, respectively. Compared with 2011, the value of forest food increased by 130%, which was the most substantial increment in Chinese non-wood forest products. Specifically, forest food could be divided into edible fungi, wild vegetables, and others. (China Forestry Bureau, 2012-2016, 2017a, 2018; China Forestry and Grassland Bureau, 2019; Zhang, 2018; Wiersum, 2017; Censkowsky, 2007). Recent surveys show that the average European household consumes about 60 kilograms of forest food annually (Lovric et al., 2020).

Compared with the traditional agricultural value chain (AVC), FFVC has less input of non-natural resources in its pre-production and production stages. It is because farmers do not use genetically modified crops, fertilizers, pesticides, and herbicides out of consideration of the ecological protection of forests, or the public's recognition of the forest industry is low, resulting in the forest industry's human capital cannot be improved (Ratnasingam et al., 2022). Traditionally, AVC could be divided into the stages of pre-production, production, processing (transportation), sales, and marketing service. To be more specific, in the pre-production stage, there are crop R&D enterprises (as breeding) and crop input enterprises (as fertilizer, pesticide, and seeds). The production stage is filled with crop planting enterprises (or farmers). The enterprises in the processing (transport) stage are typically crop processing (sorting, cutting, mixing, refining, chemical treatment) or transportation enterprises. There are wholesale or retail companies in the sales stage, and enterprises in the marketing service stage include experiential marketing service, e-commerce service, and brand-promotion service companies (Trienekens, 2011; Higgins et al., 2010). According to the results of interviews with experts, FFVC in China lacked the stages of pre-production, production, and marketing services (Hu et al., 2018; Liu et al., 2019; Zhou, 2019).

Based on a literature review and interviews with experts, FFVC may be an essential research topic of AVC, and the value-added strategies of FFVC shall consider its input-output relationship and its influencing factors, whereas the reality is disappointing (Li et al., 2016; Chen and Zhang, 2012; Chen and Song, 2008; Liu et al., 2006). On the other hand, some works of literature discussed that the input-output relationship of AVC might be impacted by the integrity and agglomeration of FFVC (Wang et al., 2016; Zhao et al., 2015; Shi et al., 2013; Wang and Zhi, 2012). Therefore, the first aim of this paper is to build a structural equation model to explore the input-output relationship of FFVC and its mediating effect on FFVC integrity and agglomeration. Second, this paper further explored the spatial dependence of FFVC’s integrity and agglomeration, and the results of the spatial econometric model could be used as policy recommendations for FFVC.

Agyeman and Ochuodho (2021) thought capital and labour endowments positively and significantly influenced forest industry structure. Assa (2018) and AFDB/OECD/UNDP (2017) argued that industrial structure and scale were the important channels between foreign direct investment and forest resources degradation. Zhang et al. (2017) and Furdychko et al. (2022) held that the influencing factors of regional industrial eco-efficiency in China included environmental regulation, technological innovation, level of economic development, and industrial structure. Dasgupta and Stiglitz (1980a) found that industrial structure and concentration, and considerable size spur inventive activity were the drivers of innovations. Teece (1996) indicated that market structure, firm boundaries (the level of integration), the structure of financial markets, and formal and informal organizational structure must be recognized as significant determinants of the rate and direction of innovation. Dasgupta 2017b and Stiglitz (1980b) proposed a relationship between Research and Development expenditure and industrial structure depends on more basic ingredients, such as technology research, demand conditions, and the nature of capital markets.

The government, enterprises, and scholars could do the marketing and promoting strategies of FFVC based on the discussion of integrity and agglomeration of FFVC. Research on FFVC integrity and agglomeration shall explore its mediating effect on the input-output relationship of FFVC by associating FFVC's various economic elements in different regions, which can be represented by the proportions of the various economic factors of FFVC in different regions. From the perspective of regional economic theory, the reasons why FFVC operators are concerned about the spatial concentration or clustering of FFVC can be concluded as follows: first, the Development Dynamics of FFVC. FFVC's operators would like to understand and make use of their regional singularities, and the successful regional clusters of FFVC would bring performance and competitiveness into its sectors with the specific locational advantages of FFVC (Morrissey and O'Donoghue, 2012; Chang, 2011; Doloreux and Shearmur, 2009; Doloreux, 2004); second, innovation Dynamics of FFVC. FFVC's innovative factors shall be interactive learning and regional externalities of cultural, economic, and institutional environments (Morrissey and O'Donoghue, 2012; Chang, 2011; Doloreux and Shearmur, 2009; Doloreux, 2004).

The contributions of this paper are as follows: (1) Based on the industrial economy, and ecological protection of FFVC, this paper explored the input-output relationship of FFVC so as to narrow the gap in empirical studies about AVC theory. (2) This paper also explored the relationship between the input, output, integrity, and agglomeration of FFVC, which may further enrich agricultural economic theory. (3) This paper is one of the first few studies discussing the integrity of FFVC and the mediating effect of agglomeration in the input-output relationship of FFVC and may, therefore, provide suggestions for policy-making. (4) This paper analyzed the spatial dependencies of FFVC in the relevant regions by their integrity or agglomeration of FFVC and provided a theoretical reference for the government to formulate differentiated forest food support policies for different regions. (5) This paper developed a structural equation model to explore the relationships among the input, output, integrity, and agglomeration of FFVC. Most AVC theories included approaches such as surveys, observation, and experimental methods and rarely used empirical models. Empirical shreds of evidence emerging from the present study may further complement agricultural economic theory.

The state-owned forest areas in northeast China are essential for more than 50% of Chinese forest areas, and they are also the primary source of forest food in China, with their output accounting for more than 98% of the output of Chinese state-owned forest areas. In northeast China, the state-owned forest areas mainly include the Jilin forest, Heilongjiang forest, and Daxinganling forest. This paper included the FFVC of these forest areas as an example to explore its input-output relationship and the mediating effect of integrity and agglomeration of FFVC and further explore the spatial dependence of integrity and agglomeration of FFVC. The rest of this paper consists of three parts. Section Two discusses the literature review and hypotheses on the inputs, outputs, integrity, and agglomeration of FFVC. Section Three describes and analyses the statistics and PLS results of the structural equations on the inputs, outputs, integrity, and agglomeration of FFVC. Section Four presents the conclusions and suggestions for future research.”

Thank you very much for indicating the policy recommendation. We have modified the policy recommendation. As ” (4) FFVC’s related activities and industries should be developed for the sustainability of forests and forestry, as the activities and industries of forest-based health preservation, forest-based carbon sink, featured economic forest, bamboo and rattan, seedlings and flowers, wild animal and plant breeding and utilization, and forest-based Internet of Things. Ma and Zheng (2017) thought the developments of the under-forest economy are important in the sustainability of forestry. Chu and Zhang (2014) used the theory of ecology-industry symbiosis to analyze the inner and outer interaction mechanism between forest ecological security and forest food security. Its results show that there is a mutually reinforcing relationship between forest ecological security and forest food security. ”

Point 2: In my opinion there is a severe lack in non considering the social environment. In particular I expect that issues like social embeddedness of foresrty, food sovereignty, affirmative policy to include women in forestry management were considered.

Response 2:

Thank you very much for indicating the social environment. We have added the future study. As ” The future study could focus on the issues of FFVC, such as its social embeddedness, food sovereignty, and affirmative policy, to include women, as Taylor and Cheng (2012) thought the social embeddedness of forestry could promote democracy, local livelihoods, and sustainable forest ecosystems. Moreno-Calles et al. (2017) thought the development of forestry should consider forestry and agricultural diversity, soil, water, cultural richness, food sovereignty, and sustainable ecosystem management. Johansson et al. (2020) discussed women professionals' conditioned openings and restraints in male-dominated forestry. Jefferson and Adhikari (2019) thought governments should promote food sovereignty and protect plant varieties as intellectual property. ”

Point 3: References in introduction should be improved.

Response 3:

Thank you very much for indicating the references of the introduction. We have modified the introduction. As ”

  1. Introduction

With an increase in governmental promotion policies and consumers' considerations of food security and nutrition, forest food has become an important economic crop, which boosts the economic sustainability of forest value chains. Therefore, the purpose of this paper is to study the sustainable development of forest food by surveying the input-output relationship of the forest food value chain (FFVC) and the factors that affect the relationship (Zhu and Zhang, 2014; Jiang and Xu, 2016; Zhang et al., 2017; Jendresen and Rasmussen, 2022). Based on a literature review, the purposes of FFVC policies include food security, poverty-reducing, and industrial up-gradation, which was concluded as the production and marketing plan, consumer participation, industrial chain management, and FFVC surveys at a regional and national scale (Agúndez et al., 2018; Food and Agriculture Organization of the United Nations, 2016). China Forestry Bureau has recognized that FFVC was the value-added sector for the forestry industry development and was conducive to developing FFVC with regional characteristics through financial subsidies (China Forestry Bureau, 2017b).

The current literature on forest food focuses on the nutritional knowledge of forest food (Bukowski and Munsell, 2018), cultural transformation (Wartman et al., 2018), nutritional benefits (Nytofte and Henriksen, 2019), and ecological restoration potential (Park and Higgs, 2018). Recently, the nutritional benefits of forests have been widely acknowledged (Gergel et al., 2020). Forest food could improve consumers’ diet quality and nutrition (Albrecht and Wiek, 2021; Tata et al., 2019), and Agúndez et al. (2018) and Gebauer et al. (2016) found there had been growing international interest in forest food and related products in FFVC’s local diet culture and tradition, health promotion, food security, species protection, and local community development. Forest foods contribute to the diversity of human diets (Hall et al., 2019; Rasmussen et al., 2019, 2020). Chamberlain (2020) and Nurhasan (2022) thought forest products are more "natural" and healthier than food produced in agriculture.

There are more than 300 types of food, pharmaceuticals, or cosmetics containing ingredients derived from baobab in Europe. Wiersum (2017) thought that the consumers in the Netherlands tend to experience the natural and cultural identity of forest food. According to Zhao et al. (2015) and Yu and Jiang (2012), forest food could be appreciated with no pesticides and no additives in the production process. Schumann et al. (2011), Venter and Witkowski (2011), Dhillion and Gustad (2004) found that human activities and the used types of land were the influencing factors of product yield from the non-timber forest in West Africa. Those human activities consisted of harvesting, seedling protection, transplanting, dispersal of seeds in the garbage, livestock grazing, and plowing, and those used types of land are nature reserves, rocky outcrops, plains, fields, and village (habitation) areas. Lacuna-Richman (2006) discovered that forest food was considered as a supplement to food instead of the source of income, and the most important factor in forest food consumption shall be the effect of the market economy.

The sustainable development of FFVC should help the sustainability of forests, for environmental and economic sustainability should be considered for the government aiming to improve the healthy development of forests and their value chains (Dalei and Gupt, 2019; Agúndez et al., 2018; Schumann et al., 2011; Venter and Witkowski, 2011; Dhillion and Gustad, 2004). For the sake of the ecological and environmental benefits of forests, numerous countries have banned forest value chains from exploiting direct economic benefits of deforestation, such as logs or wood pulp. However, in the light of sustainable development of forests, countries still encourage forest value chains to develop the indirect economic benefits of the forest, such as FFVC, eco-tourism, animal and plant breeding, and utilization. Based on previous literature, the ecological and environmental benefits of forests are becoming increasingly important, such as climate regulation, air purification, soil and water conservation, and biodiversity maintenance (Molander et al., 2022; Liu et al., 2022; Villamor et al., 2022; Andrea, 2022). Furthermore, Jendresen and Rasmussen (2022) thought the bottom wealth group has a higher frequency of forest food consumption. Durazzo et al. (2020) thought forests and trees are essential in food production. Sunderland and Vasquez (2020) believe that we must strike a balance between maintaining forest ecological diversity and developing forest industries. Elbakidze et al. (2007) believed that forests were of environmental importance for biodiversity, the global carbon cycle, and the international food trade. Graham (2004) found that structure and diversity of forest-related ecosystems may depend on the forest itself.

The commercial logging of state-owned natural forests has been forbidden in northeast China since April 2014. Such an economic development model is in line with the national industrial policy, which vigorously promotes green alternative industries such as forest food, ecological tourism, planting and breeding, and forest pharmaceutical industries. Shackleton et al. (2011) discussed the use, management, and marketing of forest food. Kusters et al. (2006) and Sunderlin et al. (2005) found that forest food production could contribute to forest conservation and livelihood improvement. Steel et al. (2022) and Asprilla-Perea et al. (2019) thought forest food generates community income. Makarov (2019) suggested that special attention should be paid to organizing and strengthening the cultivation of forest fruits and medicinal plants to cultivate new varieties with high yields.

According to various international and national unions and regulations, forest food refers to the edible plants or parts growing naturally in forests. Based on the statistical classification of forest food in the China Forestry Statistics Yearbook, the Chinese forest coverage rate was 22.96% by the end of 2018. The output and value of forest food were 678,600 tons and RMB 10.723 billion, respectively. Compared with 2011, the value of forest food increased by 130%, which was the most substantial increment in Chinese non-wood forest products. Specifically, forest food could be divided into edible fungi, wild vegetables, and others. (China Forestry Bureau, 2012-2016, 2017a, 2018; China Forestry and Grassland Bureau, 2019; Zhang, 2018; Wiersum, 2017; Censkowsky, 2007). Recent surveys show that the average European household consumes about 60 kilograms of forest food annually (Lovric et al., 2020).

Compared with the traditional agricultural value chain (AVC), FFVC has less input of non-natural resources in its pre-production and production stages. It is because farmers do not use genetically modified crops, fertilizers, pesticides, and herbicides out of consideration of the ecological protection of forests, or the public's recognition of the forest industry is low, resulting in the forest industry's human capital cannot be improved (Ratnasingam et al., 2022). Traditionally, AVC could be divided into the stages of pre-production, production, processing (transportation), sales, and marketing service. To be more specific, in the pre-production stage, there are crop R&D enterprises (as breeding) and crop input enterprises (as fertilizer, pesticide, and seeds). The production stage is filled with crop planting enterprises (or farmers). The enterprises in the processing (transport) stage are typically crop processing (sorting, cutting, mixing, refining, chemical treatment) or transportation enterprises. There are wholesale or retail companies in the sales stage, and enterprises in the marketing service stage include experiential marketing service, e-commerce service, and brand-promotion service companies (Trienekens, 2011; Higgins et al., 2010). According to the results of interviews with experts, FFVC in China lacked the stages of pre-production, production, and marketing services (Hu et al., 2018; Liu et al., 2019; Zhou, 2019).

Based on a literature review and interviews with experts, FFVC may be an essential research topic of AVC, and the value-added strategies of FFVC shall consider its input-output relationship and its influencing factors, whereas the reality is disappointing (Li et al., 2016; Chen and Zhang, 2012; Chen and Song, 2008; Liu et al., 2006). On the other hand, some works of literature discussed that the input-output relationship of AVC might be impacted by the integrity and agglomeration of FFVC (Wang et al., 2016; Zhao et al., 2015; Shi et al., 2013; Wang and Zhi, 2012). Therefore, the first aim of this paper is to build a structural equation model to explore the input-output relationship of FFVC and its mediating effect on FFVC integrity and agglomeration. Second, this paper further explored the spatial dependence of FFVC’s integrity and agglomeration, and the results of the spatial econometric model could be used as policy recommendations for FFVC.

Agyeman and Ochuodho (2021) thought capital and labour endowments positively and significantly influenced forest industry structure. Assa (2018) and AFDB/OECD/UNDP (2017) argued that industrial structure and scale were the important channels between foreign direct investment and forest resources degradation. Zhang et al. (2017) and Furdychko et al. (2022) held that the influencing factors of regional industrial eco-efficiency in China included environmental regulation, technological innovation, level of economic development, and industrial structure. Dasgupta and Stiglitz (1980a) found that industrial structure and concentration, and considerable size spur inventive activity were the drivers of innovations. Teece (1996) indicated that market structure, firm boundaries (the level of integration), the structure of financial markets, and formal and informal organizational structure must be recognized as significant determinants of the rate and direction of innovation. Dasgupta 2017b and Stiglitz (1980b) proposed a relationship between Research and Development expenditure and industrial structure depends on more basic ingredients, such as technology research, demand conditions, and the nature of capital markets.

The government, enterprises, and scholars could do the marketing and promoting strategies of FFVC based on the discussion of integrity and agglomeration of FFVC. Research on FFVC integrity and agglomeration shall explore its mediating effect on the input-output relationship of FFVC by associating FFVC's various economic elements in different regions, which can be represented by the proportions of the various economic factors of FFVC in different regions. From the perspective of regional economic theory, the reasons why FFVC operators are concerned about the spatial concentration or clustering of FFVC can be concluded as follows: first, the Development Dynamics of FFVC. FFVC's operators would like to understand and make use of their regional singularities, and the successful regional clusters of FFVC would bring performance and competitiveness into its sectors with the specific locational advantages of FFVC (Morrissey and O'Donoghue, 2012; Chang, 2011; Doloreux and Shearmur, 2009; Doloreux, 2004); second, innovation Dynamics of FFVC. FFVC's innovative factors shall be interactive learning and regional externalities of cultural, economic, and institutional environments (Morrissey and O'Donoghue, 2012; Chang, 2011; Doloreux and Shearmur, 2009; Doloreux, 2004). ”

Point 4: The paragraphs from line 96 to 134 need to be reformulated, it is very hard to follow the discourse.

Response 4:

Thank you very much for indicating the paragraphs from line 96 to 134. We have modified the paragraphs from line 96 to 134. As ”

The sustainable development of FFVC should help the sustainability of forests, for environmental and economic sustainability should be considered for the government aiming to improve the healthy development of forests and their value chains (Dalei and Gupt, 2019; Agúndez et al., 2018; Schumann et al., 2011; Venter and Witkowski, 2011; Dhillion and Gustad, 2004). For the sake of the ecological and environmental benefits of forests, numerous countries have banned forest value chains from exploiting direct economic benefits of deforestation, such as logs or wood pulp. However, in the light of sustainable development of forests, countries still encourage forest value chains to develop the indirect economic benefits of the forest, such as FFVC, eco-tourism, animal and plant breeding, and utilization. Based on previous literature, the ecological and environmental benefits of forests are becoming increasingly important, such as climate regulation, air purification, soil and water conservation, and biodiversity maintenance (Molander et al., 2022; Liu et al., 2022; Villamor et al., 2022; Andrea, 2022). Furthermore, Jendresen and Rasmussen (2022) thought the bottom wealth group has a higher frequency of forest food consumption. Durazzo et al. (2020) thought forests and trees are essential in food production. Sunderland and Vasquez (2020) believe that we must strike a balance between maintaining forest ecological diversity and developing forest industries. Elbakidze et al. (2007) believed that forests were of environmental importance for biodiversity, the global carbon cycle, and the international food trade. Graham (2004) found that structure and diversity of forest-related ecosystems may depend on the forest itself.

The commercial logging of state-owned natural forests has been forbidden in northeast China since April 2014. Such an economic development model is in line with the national industrial policy, which vigorously promotes green alternative industries such as forest food, ecological tourism, planting and breeding, and forest pharmaceutical industries. Shackleton et al. (2011) discussed the use, management, and marketing of forest food. Kusters et al. (2006) and Sunderlin et al. (2005) found that forest food production could contribute to forest conservation and livelihood improvement. Steel et al. (2022) and Asprilla-Perea et al. (2019) thought forest food generates community income. Makarov (2019) suggested that special attention should be paid to organizing and strengthening the cultivation of forest fruits and medicinal plants to cultivate new varieties with high yields.

According to various international and national unions and regulations, forest food refers to the edible plants or parts growing naturally in forests. Based on the statistical classification of forest food in the China Forestry Statistics Yearbook, the Chinese forest coverage rate was 22.96% by the end of 2018. The output and value of forest food were 678,600 tons and RMB 10.723 billion, respectively. Compared with 2011, the value of forest food increased by 130%, which was the most substantial increment in Chinese non-wood forest products. Specifically, forest food could be divided into edible fungi, wild vegetables, and others. (China Forestry Bureau, 2012-2016, 2017a, 2018; China Forestry and Grassland Bureau, 2019; Zhang, 2018; Wiersum, 2017; Censkowsky, 2007). Recent surveys show that the average European household consumes about 60 kilograms of forest food annually (Lovric et al., 2020).

Compared with the traditional agricultural value chain (AVC), FFVC has less input of non-natural resources in its pre-production and production stages. It is because farmers do not use genetically modified crops, fertilizers, pesticides, and herbicides out of consideration of the ecological protection of forests, or the public's recognition of the forest industry is low, resulting in the forest industry's human capital cannot be improved (Ratnasingam et al., 2022). Traditionally, AVC could be divided into the stages of pre-production, production, processing (transportation), sales, and marketing service. To be more specific, in the pre-production stage, there are crop R&D enterprises (as breeding) and crop input enterprises (as fertilizer, pesticide, and seeds). The production stage is filled with crop planting enterprises (or farmers). The enterprises in the processing (transport) stage are typically crop processing (sorting, cutting, mixing, refining, chemical treatment) or transportation enterprises. There are wholesale or retail companies in the sales stage, and enterprises in the marketing service stage include experiential marketing service, e-commerce service, and brand-promotion service companies (Trienekens, 2011; Higgins et al., 2010). According to the results of interviews with experts, FFVC in China lacked the stages of pre-production, production, and marketing services (Hu et al., 2018; Liu et al., 2019; Zhou, 2019). ”

Point 5: In line 127 there is a -first- not followed by a secondo and so on.

In line 229 there is a typo and the -i- is written -I-, please correct.

Response 5:

Thank you very much for indicating the mistakes. We have corrected.

Reviewer 2 Report

Dear Authors

Despite the interesting subject, the data that was used (from 2011-2018) are outdated and publishing this data in 2022 might not be of much interest to the readers and other scholars. Maybe paying attention to significant phenomena like Covid, which might be represented in statistical yearbooks of 2020-2022 as a critical factor for policy and lifestyle changes, could improve the readability and interestingness of your manuscript.

yours 

Author Response

Response to Reviewer 2 Comments

Point: Despite the interesting subject, the data that was used (from 2011-2018) are outdated and publishing this data in 2022 might not be of much interest to the readers and other scholars. Maybe paying attention to significant phenomena like Covid, which might be represented in statistical yearbooks of 2020-2022 as a critical factor for policy and lifestyle changes, could improve the readability and interestingness of your manuscript.

Response :

Thank you very much for indicating the data. We have added the limitations. As ” The future study could pay attention to significant socio-economic phenomena like Covid-19, which might be represented in statistical yearbooks of 2020-2022 as a critical factor for policy and lifestyle changes. The limitation of data in this paper is due to the data of the research object in the China Forestry Statistical Yearbook selected in this study being only consistent from 2011 to 2018, and there are some missing data after 2018.”

Reviewer 3 Report

Improve the wordiness and revise the manuscript. Add few latest references.

Author Response

Response to Reviewer 3 Comments

Point: Improve the wordiness and revise the manuscript. Add few latest references.

Response :

Thank you very much for your suggestions. We have improved the wordiness, revised the manuscript, and added few latest references. Please see the modified file.

Round 2

Reviewer 1 Report

With the poderouse reorganization of the introduction the article is now clearly structured and solidly grounded on the literature.

Author Response

Response to Reviewer 1 Comments

Point: With the poderouse reorganization of the introduction the article is now clearly structured and solidly grounded on the literature.

Response: Thank you very much for your review report. 

Reviewer 2 Report

Dear Authors

I see how researchers have limited data access, and yours. 

Regards

Author Response

Response to Reviewer 1 Comments

Point: I see how researchers have limited data access, and yours.

Response 1:

Thank you very much for your review report.